# Which Factors Influence the Consumption of Antibiotics in Q-Fever-Positive Dairy Farms in Northern Germany?

**DOI:** 10.3390/ani14091375

**Published:** 2024-05-03

**Authors:** Niclas Huber, Natascha Gundling, Maria Thurow, Uwe Ligges, Martina Hoedemaker

**Affiliations:** 1Clinic for Cattle, University of Veterinary Medicine Hanover, Foundation, Bischofsholer Damm 15, 30173 Hannover, Germany; 2Department of Statistics, TU Dortmund University, 44221 Dortmund, Germanyligges@statistik.tu-dortmund.de (U.L.); 3UA Ruhr, Research Center Trustworthy Data Science and Security, 44227 Dortmund, Germany

**Keywords:** *Coxiella burnetii*, vaccination, antibiotics, dairy cows

## Abstract

**Simple Summary:**

Q fever is an infection caused by the bacterium *Coxiella burnetii*. It can be transmitted from animals to humans, and the infection is usually airborne. In cattle, infection with *Coxiella burnetii* can lead to increased abortions, premature births or stillbirths, and weak calves. Since 2010, it has been possible to vaccinate cows against Q fever with the vaccine COXEVAC^®^ (Ceva Santé Animale). It was the aim of this study to examine whether the usage of this vaccine could reduce the consumption of antibiotics in Q-fever-positive dairy farms. Antimicrobial use and detailed information on herd data, nutrition, milking management, housing, and animal health were documented and evaluated on 36 vaccinated and 13 non-vaccinated dairy farms. The findings of the present study suggest that more antibiotics were used on non-vaccinated dairy farms than on vaccinated dairy farms. Neither herd size nor milk yield level influenced the consumption of antibiotics in the study herds. Floor type and udder-cluster disinfection while milking were associated with a lower and higher therapy frequency, respectively. Further studies are necessary to elucidate the cause–effect relationship between vaccination and the consumption of antibiotics on dairy farms in Northern Germany.

**Abstract:**

It was the aim of this study to examine whether the usage of the vaccine COXEVAC^®^ (Ceva Santé Animale) could reduce the consumption of antibiotics in Q-fever-positive dairy farms. Additionally, the effects of other herd-level factors on the consumption of antibiotics were investigated. A total of 36 farms with vaccination and 13 farms without vaccination participated in this longitudinal cohort study. In all herds, *Coxiella burnetii* had been directly or indirectly diagnosed. To compare the treatment frequency of antibiotics between the vaccinated group and the non-vaccinated group, the consumption of antibiotics for each farm was collected using the veterinary documents about the application and delivery of antibiotics. To gather detailed information about herd data, nutrition, milking management, housing, and animal health, the farmers were interviewed with the help of a questionnaire. The results thereof suggest that there might be an association between the vaccination against Q fever and a reduced consumption of antibiotics. Neither herd size nor milk yield level influenced the consumption of antibiotics in the study herds. Type of flooring and udder-cluster disinfection while milking were associated with a lower and higher therapy frequency, respectively. Further studies are necessary to elucidate the cause–effect relationship between vaccination and the consumption of antibiotics.

## 1. Introduction

The gram-negative pathogen that causes Q fever, *Coxiella burnetii*, is found almost everywhere in the world. It is a zoonotic pathogen that, in addition to sheep, goats, cattle, and humans, has been detected in many other animal species, such as various rodents, wild ruminants, cats, dogs, horses, pigs, marsupials, reptiles, birds, and arthropods, especially ticks [1,2]. Although wild rodents are an important reservoir for the pathogen, domestic ruminants (sheep, goats, and cattle) are the most common source of human infection [3]. The main route of infection for humans and animals is via inhalation of contaminated aerosols [4,5]. *C. burnetii* is highly infectious. In humans, the dose required for a 50% probability of infection is approximately 15 microorganisms, and one inhaled microorganism of *C. burneti* can cause infection in 5% of the exposed population [6]. In the Dutch human epidemic from 2007 to 2009, with more than 3500 reported cases, sheep and goat herds were particularly important [7]. However, infection can be asymptomatic and only seroconversion can be detected. Other people developed an acute flu-like syndrome, which can be mild and is usually not diagnosed as Q fever. Two to five percent of patients had severe complications requiring hospitalization [8,9,10]. Those patients were diagnosed with severe atypical pneumonia, hepatitis, meningoencephalitis, or myocarditis [1,8,11]. There is a particular risk of infection of humans, cattle, and other susceptible animals at the time of parturition of an infected animal due to the massive shedding of the pathogen with the amniotic fluid and the placental membranes. However, *Coxiella burnetii* is also excreted by infected animals in feces, urine, semen, and milk [12]. In cattle, *C. burnetii* can lead to increased abortions, premature births or stillbirths, and weak calves [9,13]. However, there is no solid evidence to support the hypothesis that *C. burnetii* causes disorders such as subfertility, endometritis/metritis, or retained fetal membranes. There may be an association between *C. burnetii* and subclinical mastitis in dairy cattle [14]. Since 2010, it has been possible to vaccinate cows against Q fever with the vaccine COXEVAC^®^ (Ceva Santé Animale). Vaccination did not significantly prevent shedding but was significantly associated with lower bacterial load shed, thus reducing the bacterial load generated in the environment [15]. Hypothetically, a lower infection rate could diminish the occurrence of diseases and, subsequently, the necessity of antibiotic treatments. It was the aim of this study to examine whether the usage of the vaccine COXEVAC^®^ could reduce the consumption of antibiotics in Q-fever-positive dairy farms. Additionally, the effects of other herd-level factors on the consumption of antibiotics were investigated.

## 2. Materials and Methods

### 2.1. Farms and Animals

A total of 49 dairy farms from Northern Germany participated in this longitudinal retrospective cohort study. Participation in the study was on a voluntary basis. The study farms were recruited in collaboration with nine veterinary practices that routinely treat all types of animals. Data were collected from the first quarter of 2012 to the last quarter of 2018. All participating farms were located in the federal state of Lower Saxony, north of Bremen, and had at least 40 cows in lactation. Samples for Q fever were collected by the local veterinarians because of non-specific illness symptoms or suspicion of Q fever before the beginning of the study. Most of these samples were analyzed in the laboratory of the Institute for Animal Health of LUFA Nord-West in Oldenburg, Germany. The decision to vaccinate against Q fever was made by the local veterinarians and the owners of the dairy farms independently of participation in the study. Requirements for participation in the study were the detection of *Coxiella burnetiid* antigen by PCR (in the fetus, placenta, cervix swabs, or milk) or detection of *Coxiella burnetiid* antibodies in serum or the individual milk sample of at least one cow in the herd. The average herd size (all cattle) was 330 animals and ranged between 105 and 1130 animals. The main breed was German Holstein. The average number of lactating cows was 149 and ranged between 49 and 520 animals. The average milk yield was 8717 kg and ranged between 5715 and 11,363 kg. Forty-five farms (94.4%) used an upgraded mixed ration with separate supplementation of concentrates for the cows. Four farms (5.5%) fed a total mixed ration. All farms with the exception of one operation housed the lactating cows in freestall barns with cubicles. One farm had a tied stall. Twenty-two vaccinated and two non-vaccinated farms allowed lactating cows access to pasture in the summer. On 25 farms, the animals were kept indoors all year round (14 vaccinated farms and 11 non-vaccinated farms).

### 2.2. Study Design

A total of thirty-six Q-fever-vaccinated and thirteen non-vaccinated dairy farms participated in the study. Seven farms (19.4%) completed only the basic immunization (two vaccinations with COXEVAC^®^ three weeks apart). On twenty-eight (77.8%) farms, a booster vaccination was carried out every nine months. One farm (2.8%) repeated the basic immunization after 12 months. In twenty-eight farms, all animals were vaccinated with COXEVAC^®^ from the age of three months. Eight farms vaccinated cows only. The study period for each farm lasted three years. For farms with vaccination, timepoint 0 (t_0_) was the day when the farms finished the basic immunization with COXEVAC^®^. For farms without vaccination, t_0_ was four weeks after a positive diagnosis of *C. burnetii.* Period one (Period 1 = t_−1_–t_0_) was the year before t_0_. Period two (Period 2 = t_0_–t_+1_) and three (Period 3 = t_+1_–t_+2_) spanned the first and second year after t_0_. Figure 1 shows an example of the study timeline and the on-farm data collection.

The consumption of antibiotics for each farm was recorded by means of the obligatory veterinary documentation of the application and delivery of drugs. The antibiotics belonged to twelve different antibiotic classes (sulfonamides, folic acid antagonists, penicillins, cephalosporins, aminoglycosides, tetracyclines, macrolides, lincosamides, phenicols, fluoroquinolones, polypeptides, and beta-lactamase inhibitors). In general, the antibiotics were administered by the veterinarians for the treatment of bacterial infections and for antibiotic drying off. The diseases treated with antibiotics were not evaluated. To gather detailed information about herd data, nutrition, milking management, housing, and animal health, the farmers were interviewed by means of a questionnaire. The principles of good agricultural practice and findings with scientific consensus served as the basis for the selection of categories and questions for the farm questionnaire [16]. The mean therapy frequency per cow and farm (MTF) was used to compare the antibiotic consumption of the farms with and without vaccination. The MTF specifies how many days one cow (lactation number ≥ 1) in a herd is treated with one active ingredient on average [17]. To calculate the MTF, the number of used daily doses (=number of treated cows × number of treatment days x number of active ingredients) is divided by the number of cows kept per farm per time period. The initial statistical analysis was performed using SAS Enterprise Guide 7.1 [18]. Descriptive statistics were performed for all questionnaire variables and the MTF. For the categorical variables, a simple frequency determination (FREQ procedure) was performed. For continuous variables, the mean and the standard deviation were calculated (PROC MEANS). The p-values were then calculated using a *t*-test (PROC TTEST) for normally distributed variables. For non-normally distributed variables, a non-parametric simple ANOVA (PROC NPAR1WAY) was used. For categorical variables, the Chi-square test or Fisher’s exact test (PROC FREQ) was used. A significance level of α = 0.05 was chosen. Linear mixed models were calculated using R Core Team with the aim of identifying the factors that influenced MTF [19]. In the final model, the number of variables was reduced using stepwise backward variable selection based on the Akaike Information Criterion (AIC).

Selected variables after using AIC:Continuous variables:Annual herd milk yield (kg).Categorical variables-Vaccination against Q fever (farms with vaccination; farms without vaccination (reference));-Study period (Period 1 = t_−1_–t_0_, (reference); Period 2 = t_0_–t_+1_; Period 3 = t_+1_–t_+2_);-Floor type lactating cows (concrete floor (reference); slatted floor; grooved slatted floor, rubber floor);-Udder-cluster disinfection while milking (yes; no (reference); automatic milking system (AMS));-Fresh food after milking (yes; no (reference));-Vaccination of dams against neonatal diarrhea in calves (yes; no (reference)).Interactions-Vaccination against Q fever × study period;-Vaccination against Q fever × annual herd milk yield (kg);-Vaccination against Q fever × floor type;-Vaccination against Q fever × udder-cluster disinfection while milking;-Vaccination against Q fever × fresh food after milking;-Vaccination against Q fever × vaccination of pregnant cows against neonatal diarrhea in calves.

## 3. Results

All farms were asked which main problem they had at t_0_. A total of 63.8% of the vaccinated farms stated that they had problems with herd fertility, whereas only 38.4% of the control farms complained about this *(p* = 0.1123). Udder health problems were reported by 16.6% of the vaccinated farms and 7.6% of the non-vaccinated farms (*p* = 0.6577). Digital dermatitis was the most common claw disease (vaccinated farms: 63.8%; non-vaccinated farms: 92.3%, *p* = 0.2379). Table 1 shows the contact of lactating animals with other animal species.

### 3.1. Mean Therapy Frequency per Cow and Farm (MTF)

MTF revealed considerable fluctuations between the individual farms and time periods (Figure 2).

For all herds and time periods, the MTF ranged between 5.1 and 7.3 treatment days per cow and farm. The mean therapy frequencyper cow and farm for the three time periods were compared between the Q-fever-vaccinated and non-vaccinated farms (Figure 3). In Period 1, MTFs did not differ significantly between the study groups. In the non-vaccinated group, the MTF continually increased over time, whereas in the vaccinated group the MTF slightly decreased. In Period 3, the MTF of the non-vaccinated group was statistically significantly higher than the MTF of the vaccinated group.

### 3.2. Risk Factors Influencing the Mean Therapy Frequency per Cow and Farm

Table 2 shows the descriptive statistics of the herd-level risk factors selected for the final linear mixed model.

The results of the final linear mixed model (Table 3) revealed that the annual herd milk yield did not influence the consumption of antibiotics in the study herds. In Period 3, the non-vaccinated herds had a statistically significantly higher MTF than the vaccinated herds. A grooved slatted floor (*p* < 0.1) and a rubber floor (*p* < 0.05) were associated with a lower MTF. Udder-cluster disinfection while milking was associated with a higher MTF.

## 4. Discussion

The aim of the study was to analyze the relationship between Q fever vaccination (COXEVAC^®^) and the consumption of antibiotics on dairy farms. Therefore, the MTF was compared between farms with and without vaccination. Throughout all herds and time periods, the MTF ranged between 5.1 and 7.3 treatment days per cow and farm. Data on antibiotic consumption quantities are essential to monitor the impact of antimicrobial use reduction strategies on animal health [20]. Therapy frequency as an assessment parameter for antibiotic consumption on farms is a common measure [10,11]. It indicates the average number of days an animal in a flock was treated with antibiotics during a given observation period [21]. This parameter is also used in the German Medicinal Products Act. The project “VetCAb-Veterinary Consumption of Antibiotics” also recorded treatment frequencies in dairy cows for the years 2011 to 2014 [22]. The VetCAb project was initiated by the German Federal Institute for Risk Assessment (BfR). In contrast to the present study, in the VetCAb project, the treatment frequency was assessed every six months rather than 12 months. In addition, in that project, the number of cubicles of the stables from the farms was taken into account rather than the average number of cows kept per period. In the VetCAb project, the half-yearly treatment frequency for dairy cows was between 2.2 and 1.3. At 5.1 to 7.3, the MTF in the present study was higher than in the VetCAb project. This difference is possibly related to the selection of the study farms. In the VetCAb project, the selected farms took part in the survey on a voluntary basis, and it is likely that most of these farms were already aware of the responsible use of antibiotics. On the other hand, farms that expected a high level of use may have been reluctant to participate. As the focus of our study was the effect of Q-fever vaccination, the selection criteria were different from those used in the VetCAb project. In addition, the current farms already had health problems. This could explain the higher MTF on the participating farms compared to the population of dairy farms in Germany. Furthermore, seasonal differences may have had a greater impact in a six-month evaluation period than in a 12-month evaluation period.

Concerning the impact of MTF on antibiotic consumption, there was an increase in MTF from 5.82 to 7.28 on non-vaccinated farms and a decrease from 5.42 to 5.11 treatment days on the Q-fever-vaccinated farms. The difference in MTF between vaccinated and non-vaccinated farms, therefore, suggests that vaccination did indeed have an effect on antibiotic consumption. This is in line with the findings of [22], who found reduced pathogen shedding and consequently a lower rate of new infections after Q-fever vaccination. In the study by [23], the reactions of the livestock owners to the effects of vaccination with COXEVAC^®^ were almost all positive. In 84% of the cases, the owners noticed a clear improvement in the health problems that had previously occurred in their herds. The health problems that led to vaccination in the study of [23] were probably similar to the reasons that led to the decision to test herds for Q fever and the decision to vaccinate herds in the present study. In the study by [23], the reasons were described in detail. These reasons included increased abortion on 67% of the farms and fertility problems on 62% of the farms. In addition, there was a sudden drop in milk yield (55%), increased incidence of pneumonia (51%), and non-specific infections or weakness with recurrent fevers (48%). In more than 20% of the farms, there was an increased incidence of births of sick and weak calves. Those results support the hypothesis of the present study, namely that the use of antibiotics is reduced due to fewer clinical diseases. It cannot be excluded that, in addition to the pathogen-specific effect, there is also a non-specific immunostimulatory effect of vaccination. In the present study, the possible *Coxiella*-related immunosuppression, similar to that known in human patients [8], was less severe. It should be reiterated that the MTF in the period t_−1_ to t_0_ was equally high in both study groups (5.42 to 5.82). This indicates a similar health status and similar treatment management at time t_0_. This factor can be considered positive for the comparability of the farms. On the non-vaccinated farms, the lack of vaccination favors an increasing spread of *C. burnetii,* and the rate of new infections is increased, thus increasing the MTF in the non-vaccinated farms. As the infection cycle of *C. burnetii* tends to take years, as described by [24], the MTF could also slowly increase as a result.

In addition to vaccination, two other factors influencing MTF were identified in the linear mixed model: the type of floor in the alleyways and the use of udder-cluster disinfection while milking. It was found that the presence of a rubber coating on the slatted floor significantly decreased MTF. However, as only four farms used this type of flooring, it was not possible to calculate an interaction estimate, and the result should be treated with caution. Nevertheless, the slatted floor also showed a reduced treatment frequency compared to the reference of a concrete floor. On a grooved slatted floor, the effect was even greater. In conclusion, the type of floor is an important factor influencing MTF. This is in line with studies by [25,26,27], who found better claw health in animals housed on rubber floors. A concrete floor is considered to be more difficult to keep clean. Frequently the used scraping systems create a wave of manure that cows often have to wade through. This causes constant stress to the horn and skin of the foot and allows infectious claw diseases to establish [27]. One of the most common infectious claw diseases is digital dermatitis, which is often treated with antibiotic sprays [27]. Poor udder cleanliness as an indicator of overall poor barn hygiene was associated with lameness. Fecal contamination of stalls probably increased the bacterial load, which may promote infectious claw lesions [28]. It is likely that slatted floors can improve leg cleanliness. As a result, the risk of infectious claw diseases was reduced. This could lead to a lower MTF compared to concrete floors.

On the other hand, the parameter udder-cluster disinfection while milking had a strong increasing influence on the MTF. This management tool to reduce udder infections while milking has been studied for many years [29]. Recent studies show that the use of peracetic acid-based udder-cluster disinfection can lead to a reduction in teat-canal colonization [30]. In automatic milking systems (AMS), interim disinfection is state of the art and is performed fully automatically after each milking. The cleaning and disinfection methods used in AMS are mainly steam disinfection or the use of peracetic acid [31]. Contrary to expectations, the final model revealed that farms using intermediate disinfection had an increased MTF. It is assumed that these farms have a general udder health problem and that the intermediate udder-cluster disinfection was introduced as a prophylactic measure to minimize new infections during milking. Conversely, farmers with good udder health do not go to the trouble of performing intermediate disinfection. Therefore, the higher mean therapy frequency per cow on farms using udder-cluster disinfection while milking may be related to a mastitis problem.

In conclusion, it seems likely that vaccination has a positive effect on herd health. Limitations to the generalizability of the study results arise from the sample size, in particular the overrepresentation of vaccinated farms. A larger sample size would be desirable for further studies, especially for the control group. In addition, it would be interesting to observe the farms with ongoing booster vaccination for a longer period of time. Another way to determine the effect of vaccination is to divide a whole herd into vaccinated and control animals. Consequently, all farm-specific factors would then be standardized. However, a persistent circulation of the pathogen through the control animals would possibly have a confounding effect so that there would be a difference in a fully vaccinated herd.

## 5. Conclusions

The present results suggest that there may be an association between the vaccination against Q fever (COXEVAC^®^) and a reduced consumption of antibiotics. Further studies are needed to clarify this cause–effect relationship between vaccination and the consumption of antibiotics on dairy farms. The variables of floor type and udder-cluster disinfection while milking were shown to be important factors influencing the therapy frequency on Q-fever-positive dairy farms in Northern Germany.

## Figures and Tables

**Figure 1 animals-14-01375-f001:**
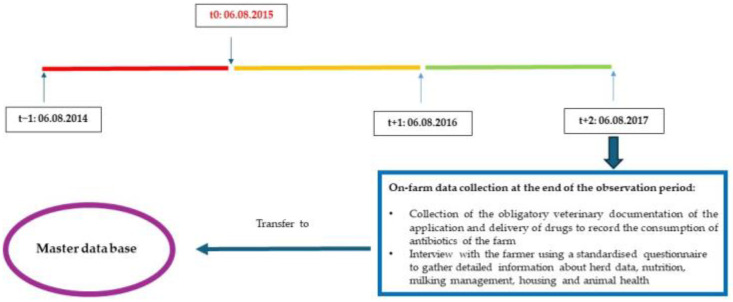
Example of the study timeline and the on-farm data collection.

**Figure 2 animals-14-01375-f002:**
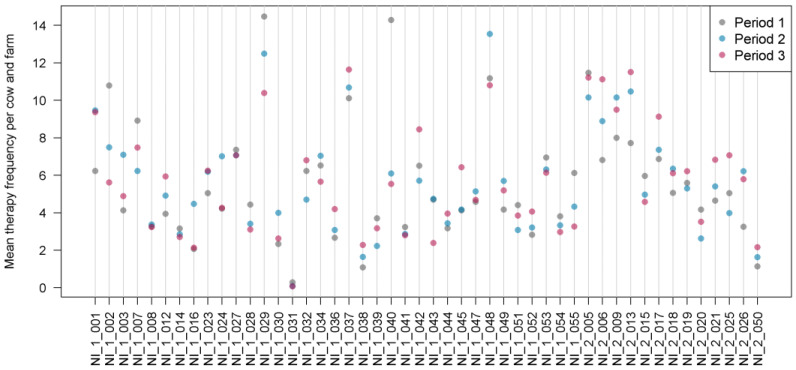
Mean therapy frequency per cow and farm in Q-fever-vaccinated and non-vaccinated dairy farms during the study periods (NI_1 = Q fever-vaccinated farms, NI_2 = non-vaccinated farms). Each dot represents a farm, and the different colors indicate different time periods.

**Figure 3 animals-14-01375-f003:**
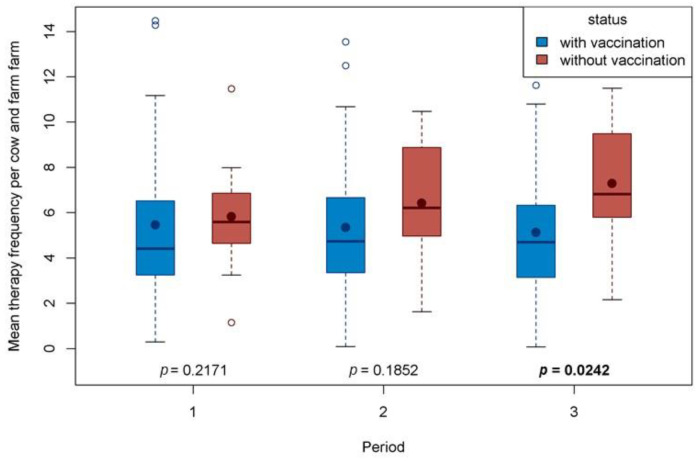
Mean therapy frequency per cow and farm compared between the study time periods in Q-fever-vaccinated and non-vaccinated dairy farms.

**Table 1 animals-14-01375-t001:** Contact of lactating cows with other animal species (multiple responses were possible, n = number of farms).

	Vaccinated Farms (n = 36)		Non-Vaccinated Farms (n = 13)	
Animal Species	n	%	n	%
**Dogs**	27	75.0	10	76.9
**Cats**	31	86.1	10	76.9
**Sheep**	4	11.1	0	0.0
**Goats**	3	8.3	0	0.0
**Horses**	3	8.3	1	7.6
**Harmful rodents**	36	100.0	13	100.0

**Table 2 animals-14-01375-t002:** Descriptive statistics of the herd-level risk factors selected for the final linear mixed model (ref. = reference, SD = standard deviation, P = period, AMS = automatic milking system).

Risk Factor	Level	Vaccinated Farms	n = 36	Non-Vaccinated Farms	n = 13
Continuous	Variables	Mean	SD	Mean	SD
	Period 1 (ref.)	5.4	3.3	5.8	2.5
**Period**	Period 2	5.3	2.9	6.4	2.9
	Period 3	5.1	2.6	7.3	3.0
**Annual herd**	Period 1 (ref.)	8579	1278	8816	990
**milk yield (kg)**	Period 2	8632	1240	9018	799
	Period 3	8655	1251	9108	520
**Categorical**	**Variables**	**n**	**%**	**n**	**%**
	Concrete floor (ref.)	3	8.3	3	23.0
**Floor type**	Slatted floor	18	50.0	7	53.8
**lactating cows**	Grooved slatted floor	11	30.5	3	23.0
	Rubber floor	4	11.1	0	0.0
**Udder-cluster**	No (ref.)	20	55.5	9	69.2
**disinfection while**	Yes	9	25.0	4	30.7
**milking**	AMS	7	19.4	0	0.0
**Fresh food after**	No (ref.)	18	50.0	5	38.5
**milking**	Yes	18	50.0	8	61.5
**Dam vaccination**	No (ref.)	21	58.3	8	61.5
**against neonatal** **diarrhea in calves**	Yes	15	41.6	5	38.4

**Table 3 animals-14-01375-t003:** Results of the final linear mixed model of risk factors influencing the mean therapy frequency per cow and farm (ref. = reference, SE = standard error, P = period, AMS = automatic milking system).

Risk Factor	Level	Estimate	SE	*p*-Value
**Intercept**		6.253	6.042	0.3026
**Vaccination**	No (ref.)			
	Yes	−3.457	6.517	0.5967
	Period 1 (ref.)			
**Period**	Period 2	5.871 × 10^−1^	5.430 × 10^−1^	0.2825
	Period 3	1.446	5.589 × 10^−1^	**0.0112**
**Annual herd** **milk yield (kg)**		4.882 × 10^−5^	6.254 × 10^−4^	0.9379
	Concrete floor (ref.)			
**Floor type**	Slatted floor	−2.317	1.531	0.1404
**lactating cows**	Grooved slatted floor	−3.916	2.310	0.0981
	Rubber floor	−3.623	1.516	**0.0235**
**Udder-cluster**	No (ref.)			
**disinfection while**	Yes	3.700	1.325	**0.0089**
**milking**	AMS	4.125 × 10^−1^	9.936 × 10^−1^	0.6810
**Fresh food after**	No (ref.)			
**milking**	Yes	−8.356 × 10^−1^	1.285	0.5205
**Dam vaccination**	No (ref.)			
**against diarrhea**	Yes	1.732	1.379	0.2192
**Interactions**				
**Vaccination × P 2**	Vaccination × P 2	−7.353 × 10^−1^	6.314 × 10^−1^	0.2473
**Vaccination × P 3**	Vaccination × P 3	−1.818	6.451 × 10^−1^	**0.0059**
**Vaccination × annual**	Vaccination × Annual			
**herd milk yield (kg)**	herd milk yield	5.864 × 10^−4^	6.700 × 10^−4^	0.3831
**Vaccination ×**	Slatted floor	−1.221	1.950 × 10^0^	0.5359
**floor type**	Grooved slatted floor	−9.380 × 10^−3^	2.628	0.9971
**Vaccination ×**				
**cluster disinfection**	Yes	−2.233	1.598	0.1725
**Vaccination × fresh**				
**food after milking**	Yes	−6.992 × 10^−1^	1.523	0.6496
**Vaccination ×**				
**dam vaccination**	Yes	2.954 × 10^−1^	1.568	0.8518

## Data Availability

Data are available on request to authors.

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
