# Peer review of "Which Factors Influence the Consumption of Antibiotics in Q-Fever-Positive Dairy Farms in Northern Germany?"

_animals, 2024, doi:10.3390/ani14091375_

Round 1

Reviewer 1 Report

Comments and Suggestions for Authors

In the manuscript titled "Which factors influence the consumption of antibiotics in  Q-fever-positive dairy farms in Northern Germany?", the authors examined whether the usage of the vaccine COXEVAC® 26 (Ceva Santé Animale) could reduce the consumption of antibiotics in Q-fever-positive dairy farms. They also investigated the effects of other herd-level factors on the consumption of antibiotics. Please see my comments and suggestions below:

1. Please proofread the entire manuscript for grammatical errors and correct them.

2. In the introduction, please mention the common symptoms of Q fever in humans. Additionally, it might be worth mentioning that C. burnetii is highly infectious, and a low dose of as few as ten bacteria can make humans sick.

3. Please consider using a tabular/figure format to explain your study design

4. Did the same species of cows appear across all farms in this study? Additionally, does C. burnetii similarly infect these species?

5. Lines 96 and 97: Please use either numerical digits or spell out numbers consistently.

6. In line 97, do you mean 29 farms vaccinated their cows only once? If yes, please edit the sentence.

7. The authors don't specify which antibiotics were used in these farms. Also, could the authors clarify if the antibiotics used in these farms were exclusively for treating Q-fever?

8. The farms that treated their cows with antibiotics: Were these cows tested to be infected with C. burnetii?

9. Discussion: In the second paragraph of the discussion section, please correct me if I am mistaken, but I interpret the authors' point as suggesting that poor udder cleanliness may result in claw disease, leading to antibiotic treatment for Q-fever. If this is the case, doesn't this seem to be non-specific?

Comments on the Quality of English Language

Please proofread the entire manuscript for grammatical errors and correct them.

Author Response

Reviewer 1

In the manuscript titled "Which factors influence the consumption of antibiotics in  Q-fever-positive dairy farms in Northern Germany?", the authors examined whether the usage of the vaccine COXEVAC® 26 (Ceva Santé Animale) could reduce the consumption of antibiotics in Q-fever-positive dairy farms. They also investigated the effects of other herd-level factors on the consumption of antibiotics. Please see my comments and suggestions below:

Thank you very much for your detailed comments. I have tried to implement all comments as accurately as possible.

  1. Please proofread the entire manuscript for grammatical errors and correct them.

=> The entire manuscript was corrected concerning the english language by a native speaker.

  1. In the introduction, please mention the common symptoms of Q fever in humans. Additionally, it might be worth mentioning that C. burnetii is highly infectious, and a low dose of as few as ten bacteria can make humans sick.

=> I have added this informations to the introduction.

  1. Please consider using a tabular/figure format to explain your study design

=> A figure explaining the study timeline and the on-farm data collection was added to the text.

  1. Did the same species of cows appear across all farms in this study? Additionally, does C. burnetii similarly infect these species?

=> All the cows are Bos taurus taurus. The main breed was German Holstein.

  1. Lines 96 and 97: Please use either numerical digits or spell out numbers consistently.

=> corrected

  1. In line 97, do you mean 29 farms vaccinated their cows only once? If yes, please edit the sentence.

=> Detailed information on the vaccination policy of the vaccinated farms has been added to the text.

  1. The authors don't specify which antibiotics were used in these farms. Also, could the authors clarify if the antibiotics used in these farms were exclusively for treating Q-fever?

=> Detailed information on the antibiotics has been added to the text. The antibiotics were administered by the veterinarians for the treatment of bacterial infections and for antibiotic drying off. The diseases treated with antibiotics were not evaluated. Only the number of cows treated, the number of treatment days and the number of active ingredients were documented to calculate the mean therapy frequency per cow and farm (MTF).

  1. The farms that treated their cows with antibiotics: Were these cows tested to be infected with C. burnetii?

=> Individual animals treated with antibiotics were generally not tested for Q fever.

  1. Discussion: In the second paragraph of the discussion section, please correct me if I am mistaken, but I interpret the authors' point as suggesting that poor udder cleanliness may result in claw disease, leading to antibiotic treatment for Q-fever. If this is the case, doesn't this seem to be non-specific?

=> More details about the findings of Oehm et al. 2022 were added in the discussion. Poor udder cleanliness as an indicative of overall poor barn hygiene was associated with lameness. Faecal contamination of stalls probably increased the bacterial load, which may promote infectious claw lesions which were often treated with antibiotic sprays.

Reviewer 2 Report

Comments and Suggestions for Authors

The study looks at possible factors influencing the consumption of antibiotics in Q-fever positive farms. The main monitored factor was preventive vaccination. The design of the study is not optimal, the observed farms differ in more factors than just vaccination (rubber floor and automatic milking system appeared only on vaccinating farms). Vaccinating and non-vaccinating farms were represented in different numbers (n = 36 vs. 13). The evaluated factor is the "mean therapy frequency per cow and farm" factor, but the authors do not provide any details regarding the type of disease or the antibiotics used. It would probably be advisable to monitor the occurrence of infections caused by C. burnetii in individual farms and thus obtain data on the effectiveness of vaccination. The presented results are not very unambiguous and conclusive, the authors themselves present them cautiously with the phrases "there may be an association..." and "further studies are needed...". Overall, the manuscript is relatively brief and works only with a very limited amount of literature (27 citations). Rather than an "article" publication, I would personally recommend reworking the work into a more concise "communication" publication type.

Author Response

  • Thank you for your comments.

„The design of the study is not optimal, the observed farms differ in more factors than just vaccination (rubber floor and automatic milking system appeared only on vaccinating farms). Vaccinating and non-vaccinating farms were represented in different numbers (n = 36 vs. 13).“

  • The chosen study design in the form of a longitudinal retrospective cohort study enabled the underlying research question (Kreienbrock et al., 2012). The advantage was that a significant proportion of the farms had already been vaccinated. In addition, a very large number of vaccinated animals (> 5000) could be observed, which, in the form of a randomised controlled trial, would certainly have been at considerable financial cost. A disadvantage of this approach was a greater inhomogeneity of farm structures than would be the case with selected experimental farms. Nevertheless, 22.331 individual records were processed. The number of farms participating in the study was significantly influenced by two factors. Firstly, the selection criteria, which often resulted in control farms being excluded from the study. Additionally, when potential control farms were Q-fever antibody positive at the individual animal level, this often resulted in the need for further antigen diagnostics, as the Animal Disease Fund of Lower Saxony only provides financial support in the case of positive antigen detection in an animal on the farm (Beihilfesatzung 2015). Often, these farms had carried out the state-supported Q fever vaccination and were therefore no longer suitable as control farms. The second factor was the milk price crisis in 2016 in Germany (Bundeszentrale für politische Bildung, 2016). At that time, many farm managers refused to participate in the study because they did not know whether they would be able to continue running the farm in the following years. Others were mentally unable to support the study. During the course of the study, five farms had to be removed from the study due to closure. This resulted in a over-representation of vaccination farms (36) compared to control farms (13).

„The evaluated factor is the "mean therapy frequency per cow and farm" factor, but the authors do not provide any details regarding the type of disease or the antibiotics used. It would probably be advisable to monitor the occurrence of infections caused by C. burnetii in individual farms and thus obtain data on the effectiveness of vaccination.“

  • Our study focused on the effect of vaccination on antibiotic consumption and not on the effect of vaccination on disease incidence at herd level. As mentioned in Köchle et al. (2024) „The use of antimicrobials in dairy cows and other livestock contributes to the emergence and spread of antimicrobial resistance, which is a major threat to human and animal health (Loo et al., 2019; Abdelfattah et al., 2021; CDC, 2022). The World Health Organization (WHO) adopted a global action plan on antimicrobial resistance in 2015 (WHO, 2015). There are significant differences in the methods used to quantify antimicrobial use (Ferreira, 2017; Umair et al., 2021), which is a challenge for direct comparisons between farms and countries. Efforts are underway in the European Union to harmonise the recording and reporting of metrics (Ferreira, 2017). The defined daily dose and defined course dose methodologies have been proposed by the European Surveillance of Veterinary Antimicrobial Consumption project initiated by the European Medicines Agency (EMA) (EMA, 2013).“ Treatment frequency is also a commonly used parameter to assess antibiotic consumption on farms (van Rennings et al., 2013; Hemme et al., 2016). We therefore decided to investigate the effect of Q fever vaccination on this parameter.

„The presented results are not very unambiguous and conclusive, the authors themselves present them cautiously with the phrases "there may be an association..." and "further studies are needed...".“

  • It is good scientific practice to point out the limitations of a study. Therefore, in order to provide the best possible information to the reader, the conclusions are cautiously formulated. Limitations in the possible generalisation of the study results arise from the sample size, in particular the over-representation of vaccination farms, and possible factors on farm level influencing antibiotic consumption that had not been recorded. Also a larger sample size would be desirable for further studies, especially for the control group. In addition, a longer observation of farms with ongoing booster vaccination and a careful documentation and evaluation of the diagnoses treated with antibiotics would also be of interest. For these reasons, further studies would be useful.

„Overall, the manuscript is relatively brief and works only with a very limited amount of literature (27 citations). Rather than an "article" publication, I would personally recommend reworking the work into a more concise "communication" publication type.“

  • The revised version of the manuscript contains 31 citations. Other references can be added if required.

References

Abdelfattah, E. M., P. S. Ekong, E. Okello, T. Chamchoy, B. M. Karle, R. A. Black, D. Sheedy, W. R. ElAshmawy, D. R. Williams, D. Califano, L. F. D. Tovar, J. Ongom, T. W. Lehenbauer, B. A. Byrne, and S. S. Aly. 2021. Epidemiology of antimicrobial resistance (AMR) on California dairies: descriptive and cluster analyses of AMR phenotype of fecal commensal bacteria isolated from adult cows. PeerJ 9:e11108. https://doi.org/10.7717/peerj.11108.

Bekoe, M. Saravanan, R.K. Adosraku, and P.K. Ramkumar, ed. IntechOpen, London, UK.

Bundeszentrale für politische Bildung. Milchpreis auf historischem Tief. 2016, https://www.bpb.de/politik/hintergrund-aktuell/228385/milchpreis-auf historischem-tief

CDC. 2022. Antibiotic resistance and NARMS surveillance. Accessed December 25, 2023. https://www.cdc.gov/narms/faq.html.

EMA (European Medicines Agency). 2013. Revised ESVAC reflection paper on collecting data on consumption of antimicrobial agents per animal species, on technical units of measurement and indicators for reporting consumption of antimicrobial agents in animals. Accessed April 24, 2023. https://www.ema.europa.eu/ en/documents/scientific-guideline/revised-european-surveillance -veterinary-antimicrobial-consumption-esvac-reflection-paper -collecting_en.pdf.

Ferreira, J. P. 2017. Why antibiotic use data in animals needs to be collected and how this can be facilitated. Front. Vet. Sci. 4:213. https://doi.org/10.3389/fvets.2017.00213

Hemme, M.; van Rennings, L.; Hartmann, M.; von Münchhausen, C.; Käsbohrer, A.; Kreienbrock, L. Antibiotikaeinsatz in Der Nutztierhaltung in Deutschland: Erste Ergebnisse Zu Zeitlichen Trends Im Wissenschaftlichen Projekt „Vetcab-Sentinel“. Dtsch. Tierärztebl. 2016, 4, 516-20.

Köchle, B., Gosselin, V. B., Schnidrig, G. A., Becker, J. (2024). Associations of Swiss national reporting system's antimicrobial use data and management practices in dairy cows on tie stall farms. Journal of Dairy Science.

Kreienbrock, L.; Pigeot, I.; Ahrens, W. Epidemiologische Methoden. 5th ed., Spektrum Akademischer Verlag: Heidelberg, Germany, 2012, http://doi.org/10.1007/978-3-8274-2334-4

Loo, E., K. S. Lai, and R. Mansor. 2019. Antimicrobial usage and resistance in dairy cattle production. Pages: 1–10 in Veterinary Medicine and Pharmaceuticals. S.O.

Ministerium für Ernährung, Landwirtschaft und Verbraucherschutz Niedersachsen, Niedersächsische Tierseuchenkasse, Satzung über die Gewährung von Beihilfen, 2015, https://www.ndstsk.de/15_beihilfen.html

Umair, M., M. Mohsin, U. W. Sönksen, T. R. Walsh, L. Kreienbrock, and R. Laxminarayan. 2021. Measuring antimicrobial use needs global harmonization. Glob. Chall. 5:2100017. https://doi.org/10 .1002/gch2.202100017.

van Rennings, L.; Merle, R.; von Münchhausen, C.; Stahl, J.; Honscha, W.; Käsbohrer, A.; Kreienbrock, L. Variables Describing the Use of Antibiotics in Food-Producing Animals. Berl. Munch Tierarztl. Wochenschr. 2013, 126, 297-309.

WHO. 2015. Global action plan on antimicrobial resistance. Accessed April 24, 2023. https://www.who.int/publications/i/item/ 9789241509763

Round 2

Reviewer 2 Report

Comments and Suggestions for Authors

The changes made in the revision are more of a cosmetic nature. The previously unmentioned differences in the vaccination schedule for the group previously listed as "vaccinated" introduce additional variability and reduce the informative value of the results. The information that antibiotics belonged to all clinically used groups and were used to treat bacterial infections did not contribute much to the concretization of the results. Unfortunately, I have to insist on my previous opinion that the quality of the methodology and the obtained data are insufficient for publication in (full) article format.

Author Response

(The authors gave the same response as above.)
